# *Larix* species range dynamics in Siberia since the Last Glacial captured from sedimentary ancient DNA

Luise Schulte [1,2✉], Stefano Meucci[1,2], Kathleen R. Stoof-Leichsenring [1], Tony Heitkam [3], Nicola Schmidt[3], Barbara von Hippel[1,2], Andrei A. Andreev[1,4], Bernhard Diekmann[1,4], Boris K. Biskaborn [1], Bernd Wagner[4], Martin Melles [4], Lyudmila A. Pestryakova[5], Inger G. Alsos [6], Charlotte Clarke[6,7], Konstantin V. Krutovsky [8,9,10,11,12] & Ulrike Herzschuh [1,2,13✉]

Climate change is expected to cause major shifts in boreal forests which are in vast areas of Siberia dominated by two species of the deciduous needle tree larch (*Larix*). The species differ markedly in their ecosystem functions, thus shifts in their respective ranges are of global relevance. However, drivers of species distribution are not well understood, in part because paleoecological data at species level are lacking. This study tracks *Larix* species distribution in time and space using target enrichment on sedimentary ancient DNA extracts from eight lakes across Siberia. We discovered that *Larix sibirica*, presently dominating in western Siberia, likely migrated to its northern distribution area only in the Holocene at around 10,000 years before present (ka BP), and had a much wider eastern distribution around 33 ka BP. Samples dated to the Last Glacial Maximum (around 21 ka BP), consistently show genotypes of *L. gmelinii*. Our results suggest climate as a strong determinant of species distribution in *Larix* and provide temporal and spatial data for species projection in a changing climate.

[1] Alfred Wegener Institute Helmholtz Centre for Polar and Marine Research, Polar Terrestrial Environmental Systems, Potsdam, Germany. [2] Institute of Biochemistry and Biology, University of Potsdam, Potsdam, Germany. [3] Institute of Botany, Technische Universität Dresden, Dresden, Germany. [4] Institute of Geology and Mineralogy, University of Cologne, Cologne, Germany. [5] North-Eastern Federal University of Yakutsk, Institute of Natural Sciences, Yakutsk, Russia. [6] The Arctic University Museum of Norway, The Arctic University of Norway, Tromsø, Norway. [7] University of Southampton, School of Geography and Environmental Science, Southampton, UK. [8] Department of Forest Genetics and Forest Tree Breeding, George-August University of Göttingen, Göttingen, Germany. [9] Center for Integrated Breeding Research, Georg-August University of Göttingen, Göttingen, Germany. [10] Laboratory of Population Genetics, N. I. Vavilov Institute of General Genetics, Russian Academy of Sciences, 119333 Moscow, Russian Federation. [11] Laboratory of Forest Genomics, Genome Research and Education Center, Department of Genomics and Bioinformatics, Institute of Fundamental Biology and Biotechnology, Siberian Federal University, Krasnoyarsk, Russian Federation. [12] Forestry Faculty, G. F. Morozov Voronezh State University of Forestry and Technologies, Voronezh, Russian Federation. [13] Institute of Environmental Science and Geography, University of Potsdam, Potsdam, Germany. ✉email: luise.schulte@awi.de; ulrike.herzschuh@awi.de

Recent climate warming is strongly amplified in the high latitudes of the Northern Hemisphere[1], but strongest in the Russian Arctic[2], and is expected to cause major shifts in boreal ecosystems[3]. Today, the deciduous conifer larch (*Larix*) dominates around 81% of the vast Siberian boreal forests[4]. It provides ecosystem services as a wildlife habitat, carbon stock and permafrost stabilizer, and renders strong economic benefits[4]. Understanding larch forest dynamics in response to climate changes in the past is critical for forecasting future ecosystem changes. Despite this importance, we still lack knowledge and understanding of past distribution of *Larix* species and the major drivers constraining their current distribution.

Siberian boreal *Larix* species are currently geographically split into a western and an eastern species (Fig. 1). West Siberia, from the western Urals to the West Siberian Plain, is dominated by *Larix sibirica* Ledeb., central and eastern Siberia is dominated by *L. gmelinii* (Rupr.) Kuzen. *Larix* taxonomy is still debated, and the latter species can be further divided into two varieties, subspecies or separate species, *L. gmelinii* to the west and *L. cajanderi* to the east[5]. As the two species have similar ecological traits[5,6] and the chloroplast genomes do not show any species-specific differences or spatial structure[7], we follow the classification as only one species, *L. gmelinii*[8]. Although outwardly very similar, *L. sibirica* and *L. gmelinii* differ in their ecological properties and the ecosystem services they offer[9]. *L. gmelinii* can grow in a severe continental climate and on continuous permafrost with shallow active-layer depth and is reported to stabilize the soil, decrease permafrost thaw depth, and subsequently also the amount of carbon release[10,11]. In harsh ecological conditions, it can still grow as dwarf or shrub-like forms[5]. In contrast, *L. sibirica* is more warmth-demanding and sensitive to frozen and swamped soils[9,12], grows faster[13], and produces more seeds[5]. It is still under discussion whether historical factors such as the historical distribution of species and dispersal limitations or environmental factors such as local hydrology and habitat are more restrictive for current species distribution.

*Larix* has been a well-established forest constituent in Eurasia since more than 25 million years ago[14]. Siberian larch species, in particular *L. gmelinii*, are considered to have formed due to climate changes in the Pleistocene resulting in adaptation to increasing climate continentality[5]. The adaptation to extreme cold prompts the question of whether the species could have survived the entire last glacial period or more specifically the Last Glacial Maximum (LGM, 17–23 ka BP[15]) in northern refugial populations or whether its distributional range retreated to the south like other tree species. Scarce paleobotanical data from a pre-LGM interstadial warm phase (Marine Isotope Stage (MIS) 3 or Karginskii Interstadial) showed that *Larix* forests were relatively common in northeast Siberia[16,17]. During the LGM, Siberia was dominated by open tundra and steppe vegetation[18], but scattered *Larix* pollen and macrofossil findings hint at the survival of refugial populations also in high latitudes[19–21]. However, our knowledge about these populations is limited, especially as *Larix* pollen production and preservation are poor[22]. Furthermore, it remains unclear to what extent these refugial populations contributed to a postglacial recolonization[13,23]. A second question is whether the mode of postglacial recolonization (a spread out of high latitude refugia or migrating from the south) differs among the different larch species.

Traditional paleoecological approaches are based on fossil pollen isolated from lake sediments, peat or permafrost sections, or metabarcoding using short diagnostic metabarcodes of ancient DNA extracted from the same kind of samples[19,24,25]. These methods are powerful in reconstructing the general past vegetation composition and species abundance[26], but do not allow taxonomic resolution of *Larix* into species. Studies using modern DNA can show distributions of current populations and also infer demographic scenarios. Nevertheless, tracing populations in the past using modern genetic data remains difficult. Changes in genetic variation as a result of specific events in different times (such as population expansion or retraction) can be superimposed and only a few studies have hitherto been available to serve as ground truth estimates for eastern Siberia[23,27,28]. Recently, Schulte et al.[29] developed target enrichment of ancient *Larix* DNA by using the complete chloroplast genome of *Larix* as a bait set (hybridization oligonucleotides) on sedimentary ancient DNA (sedaDNA). They showed that this method is capable of discriminating between different *Larix* species and potentially different populations over time. Chloroplasts are a promising target for enrichment as they are present in multiple copies in almost all plant cells. Other, yet unexplored, multi-copy targets for enrichment are repetitive DNA sequences in the nuclear plant genome. Occupying about 80% of the large genomes of conifers, they are believed to be the main source of genome expansion[30].

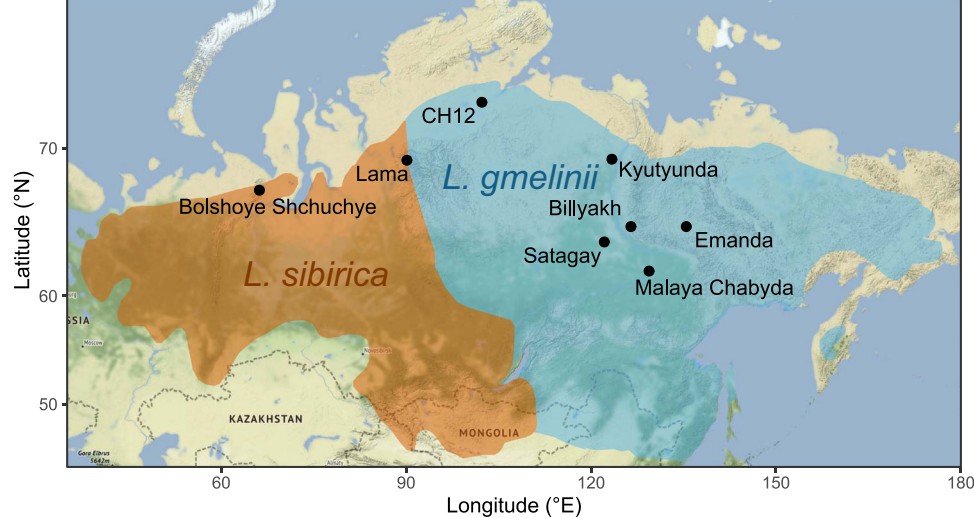

**Fig. 1 Map of study sites and current distribution of *Larix* species.** Black dots indicate positions of lakes with studied sediment cores, colors indicate current species distribution (adapted from Semerikov and Lascoux[72]). Base map is done with ggmap[73], map tiles by Stamen Design, under CC BY 3.0. Data by OpenStreetMap, under ODbL.

Satellite repeats are one class of repetitive DNA elements which form arrays of tandemly repeated non-coding DNA stretches that can affect important cell functions such as chromosome stability and cell division[31]. Among repetitive elements, satellite DNA repeats are among the fastest evolving and are hence often specific for a certain species or genus[32]. As they constitute up to 37% of some plant genomes[33], they often remain as off-target DNA sequences in target enrichment of single- or low-copy targets in modern species[34,35]. Despite their genomic abundance and their potential specificity to genus or species, enrichment of repetitive elements from sedaDNA has so far not been reported.

In this study, we applied chloroplast and nuclear target enrichment to a total of 67 sedaDNA extracts, isolated from sediment core samples from eight lakes distributed across Siberia covering the last 50,000 years (Fig. 1). The lakes were selected according to the availability of sediment records spanning the LGM and a known presence of *Larix* in the past around the lakes as indicated by other proxies from the same location such as pollen and metabarcoding. We successfully enriched sedaDNA for the *Larix* chloroplast genome and abundantly detected the *Larix* main satellite repeat in the off-target sequences. The analysis of genetic variation in the enriched sequences gave us unprecedented insights into the past distribution of species. We show that climate and local habitat such as permafrost are stronger determinants of species distribution than historical factors. We highlight that *L. sibirica* had a much wider distribution in the past; nevertheless, glacial refugial populations were dominated by *L. gmelinii*, potentially enhancing the postglacial colonization in East Siberia.

To our knowledge, this study is the first to provide large-scale species information using target enriched data of sedaDNA isolated from lake archives and the first to report the enrichment of ancient repetitive DNA.

## Results and discussion

**Chloroplast and repetitive nuclear DNA enrichment in the sedaDNA extracts**. To the best of our knowledge, we generated the first large-scale target enriched dataset using sedaDNA extracted from sediments of multiple lakes. Sequencing of two datasets produced 325.5 million (M) quality-filtered paired-end DNA sequences. The first target enriched dataset, targeting both the chloroplast and a set of nuclear genes of *Larix* on 64 sedaDNA extracts and 19 negative controls from seven lake sediment records resulted in 324 M quality-filtered paired-end sequences. The second target enriched dataset, targeting only the set of nuclear genes of *Larix* on four samples and two negative controls from an additional lake (Lake CH12) resulted in 1.5 M sequences. Quality-filtering of an additional published target enriched dataset[29], targeting the *Larix* chloroplast genome on the same CH12 samples as applied for the second dataset, added another 54 M sequences.

For the chloroplast enrichment, 390 thousand (K) sequences (1%) were classified as *Larix* at the genus or species level. The average coverage of bait regions was 19% at a mean sequence depth of 0.8. Sequencing of 19 library and extraction blank (negative control) samples resulted in 597 K paired-end sequences, of which 58% quality-filtered and deduplicated sequences remained. Of these, 38% were classified, with 0.03% of them (463 sequences) corresponding to the genus *Larix*. Negative controls from library preparation resulted in no to very few (0 to 5) sequences mapping to the *Larix* chloroplast reference genome. Negative controls from DNA extractions, which were in several cases pooled to one library, showed a low number of sequences mapped to *Larix* (0 to 94 sequences, except 237 sequences in one case). Excluding all sequences in negative

controls from the sample analysis had no impact on the patterns resulting from the analysis of sample data. Detailed results and evaluation of negative controls are included in the Supplementary Information (Fig. S5) and Supplementary Data 1 and 2. Samples of all lake records with sufficient sequence coverage showed damage patterns typical of ancient DNA (see Supplementary Data 3).

These results are comparable to the results obtained by Schulte et al.[29], where 36% of quality-filtered sequences were classified as Viridiplantae with 9% assigned to *Larix*. In contrast to[29], we raised the confidence threshold of taxonomic classification (a parameter defining the number of k-mers needed to produce a match against a taxon in the database), which drastically reduced the number of classified sequences, but increased the confidence in the analysis[36].

To analyze the enrichment obtained by the nuclear gene bait set, taxonomic classification was repeated using a plant genome database including available Pinaceae genomes. The classification resulted in 716 K sequences assigned to *Larix*, increasing the previous results by 325 K sequences. However, almost no sequences were mapped against the targeting baits (a maximum of five sequences for some samples). A closer inspection of unmapped sequences assigned to *Larix* revealed a high content of repetitive DNAs. More specifically, taxonomically classified *Larix* sequences could be assembled to EulaSat1, the most abundant satellite repeat in the nuclear genome of *Larix*[32,37]. This short repeat with a 173 bp long motif is arranged in large arrays of tandemly repeated motifs and is exclusively present in larches[32]. Analysis of modern *L. sibirica*, and *L. gmelinii* (western and eastern range) genomes reveals that EulaSat1 occurs in all species, contributing to 0.62% (*L. sibirica*), 2.52% (western range *L. gmelinii*), and 2.39% (eastern range *L. gmelinii*), of the genomes, respectively (Fig. S2). A comparison of the sequence proportions mapping to the repeat motif in the different datasets of Lake CH12 showed a specific enrichment of the repeat motif by the nuclear gene hybridization probe set (Fig. S3).

In total, 17 K sequences mapped to the repeat motif of EulaSat1. The abundance of all sequences mapped per sample is in agreement with the abundance of sequences mapped to the chloroplast genome, confirming the general history of forest development (Fig. 2). Analysis of the nucleotide frequencies in the repeat motif showed a high constancy over all samples (Fig. S4). This suggests high conservation of the EulaSat1 motif in Siberian larches over time and space. Although satellite repeats are reported to have a high sequence turnover, for larches it has been shown that repeat profiles between two geographically well-separated species—the European larch (*L. decidua*) and the Japanese larch (*L. kaempferi*)—are very similar[32]. The main satellite in all larches, EulaSat1, is believed to have greatly multiplied after the split of *Larix* from *Pseudotsuga*[32]. Given the ongoing hybridization between the three Siberian larch species, it is not surprising to find a consistent pattern of nucleotide frequencies in all samples.

Off-target sequences in target enriched datasets have already been demonstrated to be useful for the analysis of high-copy DNA such as ribosomal DNA or plastomes[34,38,39]. A recent study on five modern sedges showed that target enriched sequencing data originally targeting a set of gene exons can also be used to study the repetitive sequence fraction and even infer phylogenetic relationships based on repetitive sequence abundance[35]. Another study showed that also sequence similarities between homologous repeat motifs can be used to reconstruct phylogenetic relationships among closely related taxa[40,41]. In the case of *Larix* satellite EuLaSat1 in our study, no change in nucleotide frequencies, neither related to locations nor in time, could be detected. However, our results show that the off-target fraction in target

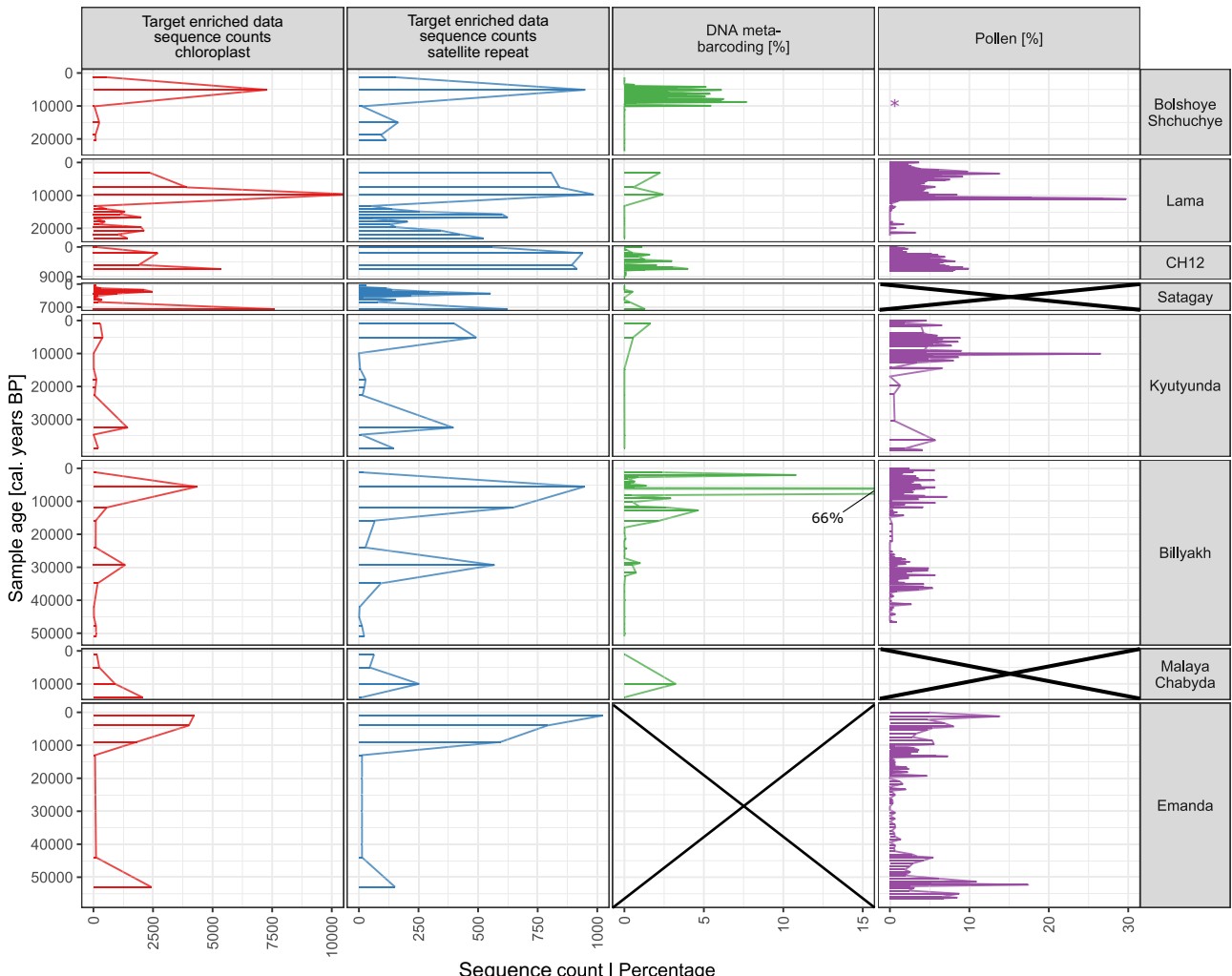

**Fig. 2 Comparison of target enrichment with available DNA metabarcoding and pollen datasets.** From left to right: *Larix*-classified sequence counts mapping to (1) the *Larix* chloroplast and (2) the EulaSat1 satellite repeat motif, (3) percentage of *Larix* counts in metabarcoding data, (4) percentage of *Larix* pollen in pollen assemblages. All data from this study, except metabarcoding data from lakes CH12[13] and Bolshoye Shchuchye[55] and all pollen data except for several samples of Lake Kyutyunda which were produced in this study[56,57,71]. Pollen data of Lake Lama and the Holocene part of Lake Kyutyunda are based on parallel sediment cores PG1111 and PG2022, respectively. No available data are marked with crosses, asterisk marks a single *Larix* pollen grain found in the Bolshoye Shchuchye sediments.

enriched sedaDNA datasets can hold valuable information and that repeat motifs in more diverse taxon groups could even be a target for enrichment. Specifically enriching for repeat motifs in sedaDNA extracts could enable the study of satellite repeat evolution as well as giving additional information on species abundance and phylogeography.

In the two target enriched datasets, sequences taxonomically classified to the genus *Larix* and mapping to the chloroplast and to the repeat sequence, respectively, show similar patterns of abundance (see Fig. 2). Compared with published metabarcoding and pollen data from the same locations, the *Larix* abundance patterns can be globally reproduced, underpinning the notion that sequence abundances in target enriched data can be used as good estimates of plant abundances. For older parts of the lake records, target enriched data show *Larix* where metabarcoding data were unable to detect a clear signal (see Fig. 2, lakes Billyakh, Bolshoye Shchuchye, Kyutyunda, and Lama). This shows that target enrichment is superior to metabarcoding when analyzing one taxonomic group in-depth, as it is less prone to errors by DNA degradation, which can impede primer binding if the

molecule becomes too short. Also, independent of age, rare taxa mostly need multiple PCR replicates to be detected by metabarcoding[42,43]. Target enrichment, however, is more sensitive in identifying one focal taxon group, as the total target length can be much larger (e.g., a complete organellar genome) than for metabarcoding, and the DNA damage patterns are put to use to authenticate ancient DNA. Also, it is limited by molecule length only by the applied threshold in the bioinformatic analysis, for which we used 30 base pairs (bp) as opposed to a minimum of 85 bp molecule length for the *Larix* metabarcoding marker (for the plant-specific trnL g/h marker[44]). Similarly, compared to traditional pollen analysis, target enrichment is more accurate at tracing a specific target group, as it is not dependent on pollen productivity. Especially in the case of *Larix*, pollen productivity is low and preservation poor, resulting in rare findings of its pollen in the sediments[22,45]. This could explain why for Lake Bolshoye Shchuchye, only a single *Larix* pollen grain was retrieved throughout the core, whereas target enrichment and metabarcoding show a strong signal in the Holocene sediments (last ~12 ka BP). Target-enriched data also records signals in MIS 2 sediments,

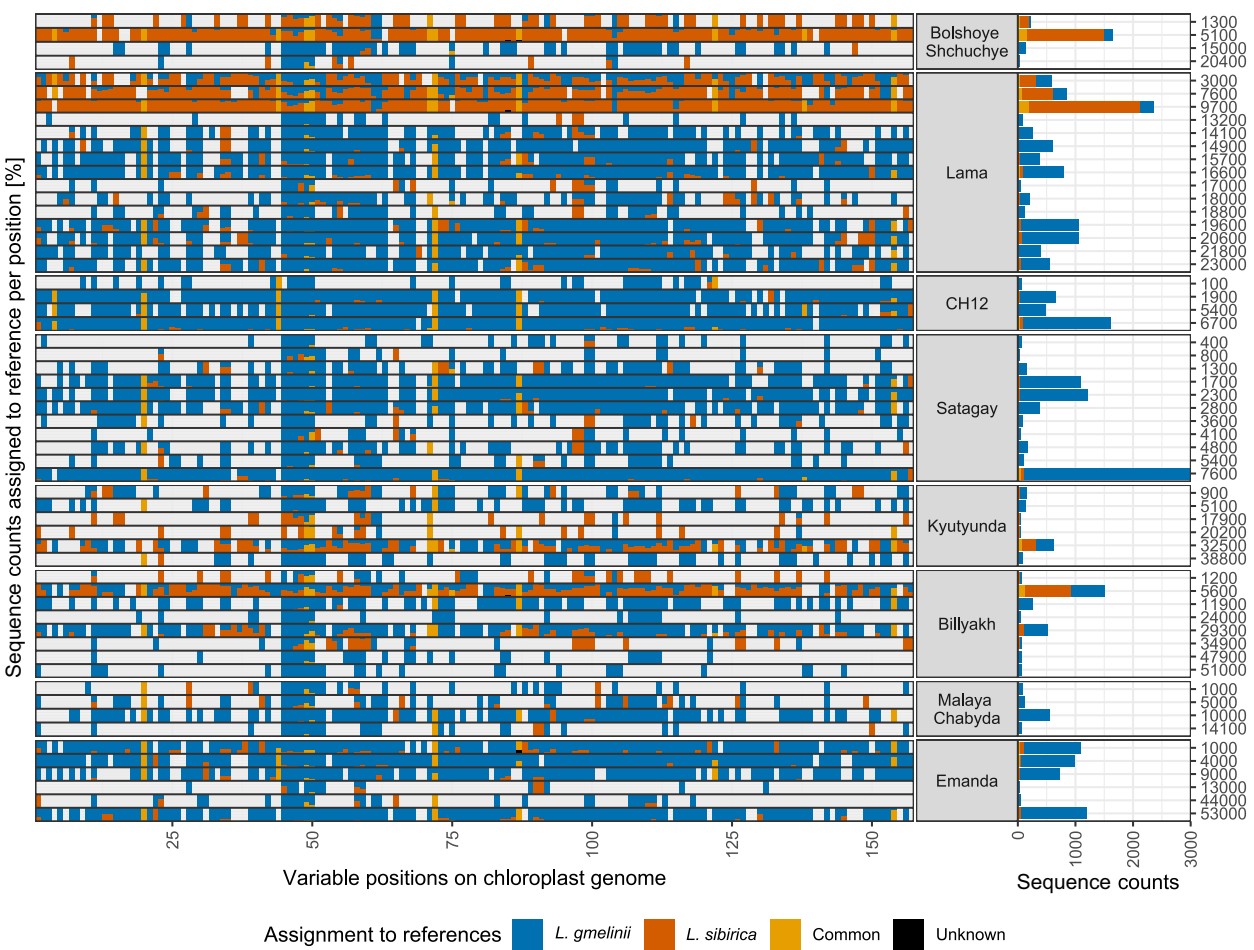

**Fig. 3 Percentage and sequence counts at variable positions along *Larix* chloroplast genome assigned to species.** Left: Alignment of *Larix*-classified DNA sequences against the chloroplast genome at the 157 variable positions between the species. For each position, the percentage of sequences assigned to a single species is displayed. Each row represents one sample named according to the calibrated age before present. Gray background indicates no coverage at the respective position. Right: Total number of sequences assigned to each of the species per sample.

however, sequence counts are extremely low, and as it is the only record, where both of the other proxies fail to report a signal, it should be interpreted with caution.

**A wider pre-glacial distribution of *L. sibirica*.** Chloroplast genomes of *L. gmelinii* and *L. sibirica* differ at 157 positions, which can be used to differentiate species in target enriched sedaDNA[29]. Here, we applied this approach to lake sediment records, which are distributed across Siberia (Fig. 1) and have time ranges back to MIS3, and thereby were able to track species composition in space and time for wide parts of the species ranges.

In lakes Billyakh and Kyutyunda, ca. 1500 km east of *L. sibirica* current range (Fig. 1), we found evidence for a wider distribution of *L. sibirica* around 32 and 34 ka BP in MIS3 (Fig. 3). Billyakh is situated in the western part of the Verkhoyansk Mountains, and Kyutyunda on the Central Siberian Plateau. Both lakes have low counts of *Larix* DNA sequences in their oldest samples dated to 51 ka BP (Billyakh) and 38 ka BP (Kyutyunda) with variants of *L. gmelinii*, but there is a sudden rise in variants attributed to *L. sibirica* at 34 ka BP (Billyakh) and 32 ka BP (Kyutyunda), which persists in the following samples, but strongly decreases in younger samples (Fig. 3). The rise in the *L. sibirica* DNA sequence variants coincides with a peak in sequence counts for Lake Kyutyunda. These signals suggest a rapid invasion of

*L. sibirica* into the ranges of *L. gmelinii* in climatically favorable times and a local depletion or extinction of *L. sibirica* during the following harsher climates. Lake Billyakh pollen data suggest a moister and warmer climate around 50–30 ka BP than in the latter part of the Last Glacial associated with the MIS3 Interstadial in Siberia[46].

Strong support for a wider pre-glacial distribution of *L. sibirica* comes from genetic analyses which show that it is genetically close to *L. olgensis*, today occurring on the Korean Peninsula and adjacent areas of China and Russia[27,47]. It is assumed that the *L. sibirica-L. olgensis* complex used to share a common range, which was disrupted and displaced when the better cold-adapted *L. gmelinii* expanded south and southwest during the more continental climatic conditions of the Pleistocene[47,48]. Furthermore, modern and ancient genetic studies suggest that the *L. sibirica* zone was recently invaded by *L. gmelinii* from the east in the hybridization zone of the species, as the climate cooled after the mid-Holocene thermal maximum[13,23]. Today, pure stands of *L. sibirica* do not form a continuous habitat, but occur in netted islands[5] and morphological features of *L. sibirica* can be found in populations of *L. gmelinii* located at least a hundred kilometers east of the closest *L. sibirica* populations[49]. Macrofossil findings of *L. sibirica* in Scandinavia dated to the early Holocene, point to the capability of rapid long-distance jump dispersal of this species[50]. Fossil *L. sibirica* cones dated to the end of the Pliocene and in the Pleistocene have also been found far east of its

current range in several river valleys including Kolyma, Aldan, and Omolon, and even in the basin of the Sea of Okhotsk[9]. These indicate long-distance seed dispersal by rivers which may also have assisted in successful establishment since the active-layer depth is deeper close to rivers[51,52]. As mentioned earlier, *L. sibirica* is sensitive to permafrost and waterlogged soils. A warmer phase with a deeper thawed layer above the permafrost could have enabled *L. sibirica* to spread and establish in regions that today are part of the geographic range of *L. gmelinii*, as *L. sibirica* is reported to have higher growth rates than *L. gmelinii*[13].

**Larix gmelinii formed northern LGM refugia across Siberia.** The possible survival of *Larix* in high latitude glacial refugia during the LGM is still under discussion[4,53] although more and more evidence is reported in favor of the existence of such refugia[17,20,21]. The question of which of the *Larix* species formed these populations has hitherto been unanswered, as both pollen and established metabarcoding markers are not able to distinguish between species in the genus *Larix*, and findings of fossilized cones identifiable to species are rare. By enriching sedaDNA extracts for chloroplast genome sequences, we are, to the best of our knowledge, for the first time, able to distinguish between *L. sibirica* and *L. gmelinii* in glacial refugial populations.

From Lake Lama, located at the western margin of the Putorana Plateau (Taymyr Peninsula), we obtained a continuous record extending from 23 ka BP to today with varying sequence counts with minima around 18–17 ka BP and 13 ka BP. All samples prior to the Holocene show variations predominantly assigned to *L. gmelinii* (Fig. 3). Our results suggest a local survival of *L. gmelinii* in the vicinity of Lake Lama throughout the LGM, which is supported by low numbers of *Larix* pollen detected through this period. Both target enriched sequence data and pollen indicate an increase from ca. 11 ka BP[54]. Sparse *Larix* pollen in the bottom part of the record could be an indication of a possible refugial population (Fig. 2; ref. [54]).

In Bolshoye Shchuchye, the westernmost lake of the study, situated in the Polar Ural Mountains, all Pleistocene samples show similarly a dominance of *L. gmelinii* sequence variations (Fig. 3). However, sequence counts for some samples are extremely low and samples from 18 and 10 ka BP had so low counts of mapped DNA sequences that none of the variable positions between the species was covered. Although sequences mapped to the satellite repeat of *Larix* also showed a Pleistocene signal, this was not repeated in pollen or metabarcoding (Fig. 2) which instead indicates a treeless arctic-alpine flora for the late Pleistocene[55,56]. Especially for the sample of 20.4 ka, *Larix* sequence counts are extremely low and new investigations would be needed to confirm a local presence of *Larix* during the LGM.

The record of Lake Billyakh situated in the western Verkhoyansk Mountain Range, likewise shows extremely low counts of sequences mapped to the reference for a range of samples with no sequences covering the studied variable sites (45, 42, and 15 ka BP, 11–56 sequences mapped to non-variable sites). However, the pollen record for the same core shows a quasi-continuous record of *Larix* with a gap only occurring during the early LGM[46] (25–22 ka BP, Fig. 2). Considering the known short-distance dispersal ability and poor preservation of *Larix* pollen, this strongly supports the presumed existence of a local glacial refugium at Lake Billyakh during that time[20]. Our samples also show a low but steady presence of *Larix* throughout the rest of the record, thus making glacial survival probable. The sample closest to the LGM (24 ka BP) indicates a clear dominance of *L. gmelinii* type variations.

The only exception to this general pattern is the record from Lake Kyutyunda, which is located on the Central Siberian Plateau

west of the Verkhoyansk Mountain Range. In this record, LGM samples have extremely low counts but show variations assigned to *L. sibirica* and not to *L. gmelinii* as in the other lakes. In addition, the preceding sample dated to the MIS3 interstadial shows *L. sibirica* variation. A possible explanation is that relics of *L. sibirica* survived during the LGM, but were unable to spread after climate warming, possibly due to genetic depletion or later local extinction. The presence of reworked sediment material can also not be excluded, as suggested by reworked pollen in the record[57].

In conclusion, our data show almost exclusively *L. gmelinii* variation for samples covering the most severe LGM climate conditions. This is in agreement with the ecological characteristics describing the species as adapted to extreme cold. In contrast to *L. sibirica*, it can grow in dwarf forms and propagate clonally and potentially survive thousands of years of adverse climatic conditions[58].

**Postglacial colonization history—differences among larch species.** Of great interest in the *Larix* history is not only the location and extent of possible high latitude glacial refugia but also if and to what extent these refugia contributed to the recolonization of Siberia after the LGM. Northern refugial populations could have functioned as kernels of postglacial population spread and recolonization, or spreading could have been driven by populations that survived in southern refugia. There are only a few studies on modern populations that report evidence for possible recolonization scenarios of *Larix*[23,27,28]. Here, we show that patterns differ between *L. sibirica* and *L. gmelinii*.

In the western part of our study region, two lakes are situated in the current distribution range of *L. sibirica* (Figs. 1, 4): Lake Bolshoye Shchuchye in the Polar Ural Mountains and Lake Lama on the Taymyr Peninsula. Despite this, both lakes show *L. gmelinii* for all Pleistocene samples, and a strong signal of *L. sibirica* variants only in the Holocene samples, with ages of 5.1 ka BP in Lake Bolshoye Shchuchye and 9.7 ka BP in Lake Lama (Fig. 3). The peak in *L. sibirica* also coincides with a peak of sequence counts in the respective sample, with a *Larix* pollen peak in Lake Lama sediments[54], and metabarcoding for Lake Bolshoye Shchuchye[55]. This points to a migration of *L. sibirica* in its current northern area of northern distribution in the course of climate warming during the early Holocene, whereas glacial refugial populations were consisting of *L. gmelinii*. Although the local survival of *L. gmelinii* around Lake Bolshoye Shchuchye remains uncertain due to extremely low sequence counts, it is clear that *L. sibirica* did not form a refugial population at this site.

A range-wide genetic study of *L. sibirica* analyzing chlorotypes and mitotypes of individuals[23] found strong indications for rapid colonization of the West Siberian Plains from populations originating from the foothills of the Sayan Mountains in the south, close to the border of Mongolia, with only limited contribution from local populations. According to our results, the local populations could have been *L. gmelinii* populations, while the rapid invasion could have been *L. sibirica*.

In the eastern range of the study region, in the current range of *L. gmelinii*, namely at lakes Emanda, Satagay, and Malaya Chabyda, genetic variations throughout the records are less pronounced. Of the three lake records, only that from Lake Emanda reaches back beyond the LGM, but with a sampling gap for the time of the LGM. Therefore, it remains uncertain whether populations survived the LGM locally, or whether they were invaded or replaced by populations coming from the south with Holocene warming. The restricted variations throughout the record, however, hint at stable populations, which is supported by scarce pollen findings (Fig. 2).

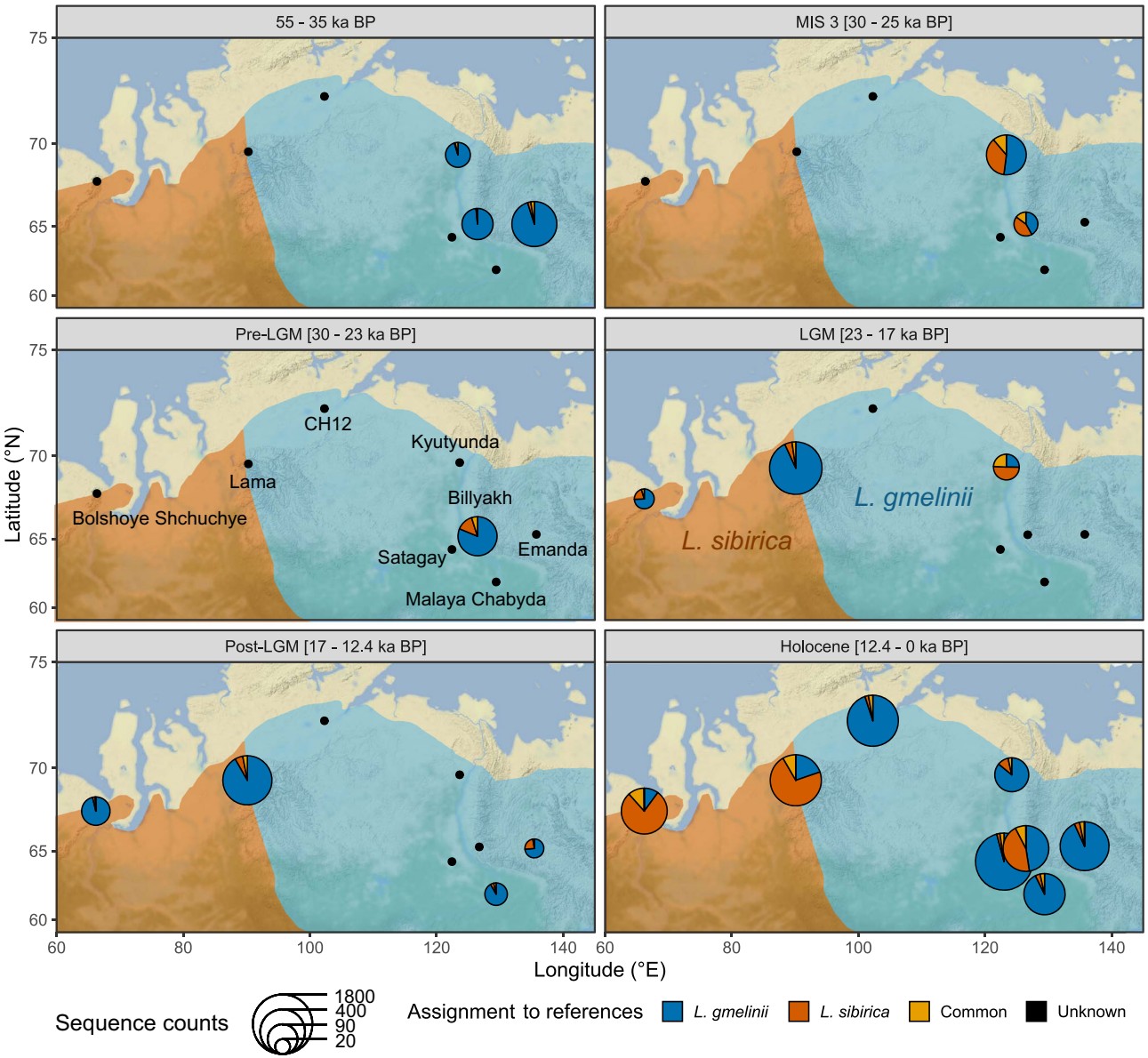

**Fig. 4 Percentage of DNA sequences assigned to references displayed on the geographical locations of the lakes investigated.** Samples in the same time frame are averaged. Lake names and current species ranges are annotated in the middle plots. Colors indicate current species distribution (adapted from Semerikov and Lascoux[72]). The base map is done with ggmap[73], map tiles by Stamen Design, under CC BY 3.0. Data by OpenStreetMap, under ODbL.

Our data suggest that postglacial recolonization of *L. sibirica* was not started from high latitude glacial refugia, but from southern populations. In contrast, northern glacial populations of *L. gmelinii* could have potentially enhanced rapid dispersal after the LGM in their current area of distribution.

**Environment likely plays a more important role than historical factors.** The current boundaries of boreal *Larix* species arranged from west to east suggest a possible strong influence of the historical species distribution on the current distribution, whereas the gradient of increasing continental climate towards the east assumes a strong influence on the environment. By tracking species distribution in the past, spanning the time of the strongly adverse climate of the LGM, we can give hitherto unprecedented insights into species distribution history.

Several lines of evidence suggest a strong influence of the environment on species distribution: (1) Signals for *L. sibirica*

appeared in its current area of distribution as late as the Holocene warming, whereas cold Pleistocene samples are dominated by *L. gmelinii* type variation; (2) in lakes far east of its modern range, signals of variation typical for *L. sibirica* coincide with peaks in sequence counts (29 ka BP, Lake Billyakh; 32 ka BP Lake Kyutyunda), which point to more forested vegetation around the lakes and consequently a more favorable climate at that time; and (3) samples dated to the severely cold LGM are dominated by variations of the *L. gmelinii* type.

This is in accordance with the different ecological characteristics described for the species. *L. sibirica* is sensitive to permafrost and only occurs outside of the zone of continuous permafrost[5]. In addition, *L. sibirica* achieves substantially higher growth rates and longer growth periods than *L. gmelinii*[9,13] and can also produce more than twice as many seeds[5]. This potentially gives *L. sibirica* the ability to quickly react to climate change and outcompete the other species when the climate becomes more favorable.

In contrast, *L. gmelinii* is adapted to extremely low soil and air temperatures and is able to grow on permafrost with very shallow thaw depths. It's distribution almost completely coincides with continuous permafrost[5], and even a restriction to permafrost areas is discussed as it does not grow well in field trials on warmer soils or where there is a small temperature gradient between air and soil[9]. Due to this ecology, *L. gmelinii* is more likely to survive in a high latitude refugium, even during the severe continental climate of the LGM, which was most probably connected to continuous permafrost of low active-layer depths.

A study combining mitochondrial barcoding on sedaDNA and a modeling approach on *Larix* distribution in the Taymyr region around Lake CH12 concluded that the distributions of *L. gmelinii* and *L. sibirica* are most strongly influenced by stand density and thus by competition between the species, with *L. gmelinii* outcompeting *L. sibirica* at high stand densities[13]. As our study includes sediment cores reaching further back in time, we see a different trend. Instead of *L. gmelinii*, it was *L. sibirica*, which dominated samples with high sequence counts, suggesting high stand density and a more favorable climate. A possible explanation for the different outcomes is the use of different organelle genomes. Epp et al.[13] used a marker representing the mitochondrial genome, which is known to introgress more rapidly and as a consequence might show a long past species history[59,60].

Our findings have potentially important implications for the projections of vegetation-climate feedback. A warming climate in conjunction with a greater permafrost thaw depth could enable the replacement of *L. gmelinii* by *L. sibirica*. In contrast to *L. gmelinii*, *L. sibirica* is not known to stabilize permafrost thus potentially further promoting permafrost thaw and with it the release of greenhouse gases, creating positive feedback on global warming[11]. On the other hand, the substantially higher growth rates of *L. sibirica* in comparison to *L. gmelinii* would increase carbon sequestration, thus mitigating global warming[13]. This shows the importance of understanding species-specific reactions to climate change, which can result in great shifts in distribution. Target enrichment applied on sedaDNA is able to reveal the impact of past climate change on populations and the increasing availability of modern reference genomes will further enhance its value of information.

## Conclusion

Our data demonstrate that Siberian larches have formed the dominant forest biomes across vast areas of Siberia since the Last Glacial. This is, to the best of our knowledge, the first time that a combination of nuclear repeats and chloroplast DNAs have been tracked backward in time to temporally resolve geospatial species distributions. Enrichment of the nuclear satellite repeat of *Larix* clearly indicated the presence of larch DNA in the sedaDNA extracts, whereas the enriched chloroplast DNA also allowed differentiation of the *Larix* species. The observed genetic variation in the target-enriched sedaDNA gave us unprecedented insights into ancient *Larix* species distribution and led to four main conclusions: (1) environment (likely permafrost) plays a more important role than historical factors for *L. sibirica*; (2) *L. sibirica* was more widely distributed during the MIS3; (3) refugial populations existed and were almost exclusively composed of *L. gmelinii*; (4) *L. sibirica* did not recolonize its current distribution area out of northern refugial populations but migrated from more southern refuges. Our high-resolution *Larix* projections highlight the importance of considering species traits in vegetation-climate-feedbacks.

## Methods

**Sample material and DNA extraction**. Samples from eight lake sediment records were included in the analysis (Supplementary Data 4). Samples from seven lake records were included in the target enrichment targeting both the chloroplast genome and a set of nuclear *Larix*, namely lakes Bolshoye Shchuchye (67.53°N and 66.18°E), Lama (69.32°N and 90.12°E), Kytuyunda (69.38°N and 123.38°E), Satagay (64.10°N and 122.15°E), Billyakh (65.17°N and 126.47°E), Malaya Chabyda (61.96°N and 129.41°E) and Emanda (65.17°N and 135.45°E). In a second target enrichment, targeting only the set of *Larix* nuclear genes, samples from Lake CH12 (72.39°N and 102,29°E) were included. DNA data of Lake CH12 samples of a target enrichment targeting the *Larix* chloroplast genome were obtained from the study of Schulte et al.[29]. Subsampling of all cores, with the exception of the core from Lake Bolshoye Shchuchye, was performed as described in Epp et al.[13] in the climate chamber of the Helmholtz Centre Potsdam—German Research Centre for Geosciences (GFZ) with the following adjustments. For each sample, the top of the sediment core half was removed twice with a sterile scalpel blade. A 1-cm thick slice of the sediment was cut out, put on a cleaned sampling plate, and sides previously touching the coring tube were cut with cleaned knives. Sampling equipment was cleaned with 5% sodium hydroxide (VWR, Germany) and DNA-ExitusPlus™ (PanReac AppliChem, Germany). Subsampling from Lake Bolshoye Shchuchye was performed in a laminar flow hood in a clean laboratory at the Centre for Geobiology and Microbiology, University of Bergen, and DNA extraction was performed in the dedicated ancient DNA facility at Tromsø University Museum as described in Clarke et al.[55]. Age models for the sediment records were adopted from refs. [46,55,57,61,62]. The age model for Lake Satagay was established in this study (see Supplementary Methods (Fig. S1) and Supplementary Data 5). DNA was extracted in a dedicated ancient DNA laboratory at Alfred Wegener Institute, Helmholtz Centre for Polar and Marine Research. Extraction was done from 1–3 g of sediment using the DNeasy PowerMax Soil Kit (Qiagen, Germany) following the manufacturer's protocol, except for adding 0.8 mg peq-GOLD Proteinase K and 100 µl of 5 M dithiothreitol (VWR, Germany) in the initial lysis and homogenization step. Further, samples were shaken for 10 min on a vortexer before overnight incubation at 56 °C in a rotating incubator. Final extracts were eluted in a 2 ml C6 solution. One milliliter of the extracts was concentrated using the GeneJET PCR Purification Kit (Thermo Scientific, USA), following the manufacturer's instruction, except for eluting in two times 25 µl, incubating for 10 min at room temperature before centrifugation. In each batch of nine samples, one negative control without sediment as input was included.

**Library preparation and bait construction**. Extracted DNA was used for single-stranded DNA library preparation following the protocol of refs. [63,64]. For each library, 30 ng of DNA extract was used. With each batch of seven libraries, one negative control (blank) library with only diethylpyrocarbonate (DEPC) purified water as input instead of DNA was included. Additional libraries were constructed for the extraction blanks. As samples from different extraction batches were included in the study, often several extraction blanks were pooled together prior to building the library (see Supplementary Data 6). Libraries were quantified with qPCR[65,66].

For the first dataset, two hybridization probe (bait) sets were used for target enrichment: (1) a set of 16 long-range PCR products covering the chloroplast genome of *Larix* excluding the ribosomal RNA regions and (2) a set of 65 candidate adaptive genes harboring in the nuclear genome of *Larix* as PCR products.

The first bait set, targeting the chloroplast genome, was produced using long-range PCR on DNA extracted from a *Larix gmelinii* individual (collected in the Botanical Gardens of the University of Potsdam, Accession CC-0POTSD-3867) as described in Schulte et al.[29]. In contrast to Schulte et al.[29], two of the 18 primer pairs covering the complete chloroplast genome[7] were left out as they cover the ribosomal RNA genes and we found that these very conserved regions cannot be unequivocally assigned to *Larix*. The omitted primer pairs are primer pair 8 (Larix_49890_F/Larix_57954_R) and primer pair 9 (Larix_57611_F/Lg_03_R).

For the second bait set, primer pairs for candidate adaptive genes found in the nuclear genome of *Larix* were selected from the literature. Specific primers for 59 candidate adaptive genes were originally designed for *Pinus taeda* by Eckert et al.[67] and tested successfully on *Larix decidua* by Mosca et al.[65]. A further six primer pairs were designed and tested on *L. sibirica* by Semerikov et al.[47], of which five single primers were modified, and two were newly designed in order to improve the primer binding affinity (see Supplementary Data 7, 8). DNA for the bait production was extracted from 80 mg of needles of a *Larix sibirica* individual sampled in 2018 from the Tazovsky District, Yamalo-Nenets Autonomous Okrug of Tyumen Oblast, Russia. The needles were transferred into impact-resistant 2 ml tubes together with two DNA-free steel beads of 5 mm diameter and ground to powder with FastPrep-24 (MP Biomedicals) for 50 s at 4 m s$^{-1}$. The DNeasy Plant Mini Kit (Qiagen) was used to isolate total genomic DNA, according to the manufacturer's protocol. The PCR products' lengths span between 169 and 683 bp (see Supplementary Data 5). The baits/amplicons were produced by PCR using 18 ng input DNA, 2.5 U Platinum™ Taq High Fidelity DNA Polymerase (Invitrogen, USA), 1x PCR buffer (Invitrogen), 0.25 mM dNTPs (Invitrogen), 0.8 mg ml$^{-1}$ BSA (VWR, Germany), 2 mM MgSO$_4$ and 0.2 µM forward and reverse primers. PCR was carried out with the following cycling conditions: 5 min initial denaturation at 94 °C, 45 cycles with 30 sec at 94 °C, 30 sec at the specific annealing temperature (see Supplementary Data 8), and 1 min at 68 °C, followed by a final extension at 10 min at 72 °C. To check for potential contamination a no template control (NTC) was carried along with each PCR extraction and treated identically

to the samples. To check successful PCR amplification, the amplicon size was checked using 1–2% agarose (Roth, Germany) gels. According to the hybridization protocol from Maricic et al.[66], bait lengths should be below 1000 bp. In order to have a uniform bait pool, five PCR products with a length higher than 1000 bp were pooled in equimolar ratios in a volume of 130 µl and sonicated using a Covaris M220 Focused-ultrasonicator (Covaris, USA) to a target peak of 450 bp (which is the mean length of the other 60 PCR products) with settings of peak incident power 50 W, duty factor 20%, cycles per burst 200 and treatment time 70 sec. The fragment size and distribution were visualized with Agilent TapeStation (D1000 ScreenTape, Agilent Technologies). The resulting fragment sizes ranged from 100 to 1000 bp with an average size of 450 bp. The sheared and non-sheared amplicons were pooled together in equimolar ratios and purified using the MinElute PCR Purification Kit (Qiagen), following the manufacturer's recommendations, and eluted in 30 µl. Blunt-ending, Adapters (Bio-T and B) ligation and Dynal™ Dynabeads™ M-270 Streptavidin (Invitrogen) ligation were conducted as described in Maricic et al.[66], with modification as described in Schulte et al.[29].

**Target enrichment by hybridization capture**. DNA libraries of the ancient lake sediment samples were pooled in equimolar amounts according to the lake of origin. Negative controls of DNA extraction and library preparation were also pooled with a fixed volume of 1 µl. According to Maricic et al.[66], ~2000 ng DNA of pooled libraries (details in Supplementary Data 6) and 500 ng of baits were used for the hybridization capture experiment. In particular, the library pools of seven lakes (see Supplementary Data 3, 6) were hybridized with 250 ng of baits targeting the chloroplast of *Larix* and 250 ng of baits targeting the 65 *Larix* candidate adaptive genes, while the Lake CH12 pool was hybridized only with 500 ng of baits covering the 65 candidate adaptive genes.

The hybridization capture experiment targeting both nuclear genes and the chloroplast genome was performed following the protocol of Maricic et al.[66] with the following modifications: (1) Two rounds of hybridization capture were performed for each pool. The input concentration, the number of amplification cycles, and the concentration of captured material after each hybridization are described in Supplementary Data 5. To prevent the binding of library molecules to the adapter sequences, which would result in off-target capture, the adapter sequences were blocked prior to the capture experiment by blocking oligonucleotides. The blocking oligos were implemented according to Schulte et al.[29]. (2) In order to reduce the tube surface and prevent the binding of magnetic beads to the tube wall during the rotation, 0.5 ml DNA LoBind® tubes (Eppendorf, Germany) were used for the hybridization capture incubation. The hybridization capture experiment of the Lake CH12 pool targeting only the 65 nuclear genes was performed as described in Schulte et al.[29], i.e., with only one round of capture in 1.5 ml DNA LoBind® tubes (Eppendorf).

**Sequencing of target enriched libraries**. The final enriched library pools targeting both nuclear genes and the chloroplast genome were pooled equimolarly to one sequencing pool of 10 nM and sequenced by Fasteris SA Sequencing Service (Geneva, Switzerland) on one SP flow-cell on an Illumina NovaSeq6000 instrument with V1.5 chemistry, 500 cycles kit and 2 × 250 bp paired-end sequencing. From the capture enriched Lake CH12 pool targeting the nuclear genes, 10 nM were sequenced by Fasteris SA Sequencing Service on one lane of an Illumina MiSeq instrument with V2 chemistry and 2 × 150 bp paired-end sequencing.

**Sequence data analysis**. To analyze the enrichment of sedaDNA libraries for *Larix* chloroplast DNA, quality-controlled DNA sequences were first classified against a chloroplast database from NCBI refseq[68] using KRAKEN2[69] with the very conservative confidence threshold of 0.8 and only sequences classified to *Larix* at genus or species level were used for further analysis. Sequences were aligned against an *L. gmelinii* chloroplast genome (NCBI GenBank accession number MK468637.1) and called variants compared to variants from all publicly available complete chloroplast genomes of *L. gmelinii*, *L. cajanderi* (NCBI GenBank accession numbers MK468630.1-36.1 & 38.1-48.1) and *L. sibirica* (NCBI GenBank accession number NC_036811.1). Sequences from the target enrichment study of Lake CH12[29,70] were included in the full analysis. Counts of species assignment per sample and position are given in Supplementary Data 11.

To analyze the enrichment of sedaDNA libraries for a set of *Larix* nuclear genes, quality-controlled DNA sequences were classified with KRAKEN2 against the plant database from NCBI RefSeq. The database was prepared with the built-in function of KRAKEN2, with a manually added *L. sibirica* genome (NCBI GenBank accession number NWUY0000000000.1) as well as available genomes of Pinaceae species (*Picea abies*, *P. glauca*, *Pinus taeda*, NCBI GenBank accession numbers GCA_900067695.1, GCA_000411955.6, and GCA_000404065.3). The classification was done using a confidence threshold of 0.8. Sequences classified as *Larix* were mapped both to the targeted bait set and the published *Larix* satellite DNAs[32] (EulaSat1-5). Nucleotide frequency per position was analyzed for the most abundant satellite repeat EuLaSat1[32]. A detailed description of the bioinformatic analysis is given in the Supplementary Information along with detailed results.

**Pollen and metabarcoding approach**. To compare enrichment of both chloroplast and repeat data, published pollen and metabarcoding records for our study sites were used. Additional metabarcoding datasets were produced for lake Billyakh and in coarse resolution for lakes Kyutyunda, Lama, Malaya Chabyda, and Satagay (Supplementary Data 9, 10). Pollen datasets from lakes Billyakh, CH12, Emanda, and Lama were retrieved from the taxonomically harmonized pollen database[71]. Pollen data from Lake Bolshoye Shchuchye were obtained from[56]. Pollen data from Lake Kyutyunda were obtained from[57] (parallel core PG2022) and additional samples were counted in this study from core PG2023 (Supplementary Data 12). Percentages of *Larix* in all used pollen and metabarcoding samples as well as counts of *Larix* sequences in target enriched data is given in Supplementary Data 13. A more detailed description of sample material and laboratory procedures is given in the Supplementary Information.

**Statistics and reproducibility**. We used 67 sediment samples originating from eight lakes across Siberia. For quality control, negative controls were included in DNA extraction, library preparation, and all PCR steps. Due to the scarcity of the sample material and the difficulty to retrieve it, no repeats were attempted. However, trends in the target enriched data were confirmed by comparison with available other proxies such as metabarcoding and pollen data. Data analysis is described in detail in the Supplementary Information in the section Supplementary Methods.

**Reporting summary**. Further information on research design is available in the Nature Research Reporting Summary linked to this article.

## Data availability

The Illumina sequence data of the target enriched dataset, targeting both the chloroplast and a set of nuclear genes of *Larix* on 64 samples and 19 negative controls from seven lake sediment records and the target enriched dataset, targeting only the set of nuclear genes of *Larix* on four samples and two negative controls from Lake CH12 are submitted to the European Nucleotide Archive under the project number PRJEB47872. Additional source data is available in the Supplementary Data.

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

## Acknowledgements

We gratefully acknowledge John-Inge Svendsen, Jan Mangerud, and Haflidi Haflidason for providing the Bolshoye Shchuchye sediment core samples. We thank Svetlana Karachurina for DNA extraction and Janine Klimke for help with DNA library preparation. We also thank Cathy Jenks for English language proofreading. This project has received funding from the European Research Council (ERC) under the European Union's Horizon 2020 Research and Innovation Program (Grant Agreement No. 772852, ERC Consolidator Grant 'Glacial Legacy') and the Initiative and Networking Fund of the Helmholtz Association.

## Author contributions

L.S. did core sampling, planned the lab work, prepared the chloroplast baits for target enrichment, and conducted bioinformatic data analysis supervised by U.H., S.M. prepared nuclear gene baits and did the target enrichment lab work, K.R.S.-L. advised lab work, T.H and N.S. analyzed repetitive elements in modern *Larix*, B.v.H. prepared libraries in the lab, A.A.A. counted Kyutyunda pollen samples for this study, B.D. coordinated fieldwork for the Billykah sediment record, B.K.B. coordinated fieldwork for Kyutyunda, Malaya Chabyda, and Satagay sediment records and did the age model for the Satagay record, B.W. coordinated fieldwork for the Emanda sediment record, M.M. coordinated fieldwork for the Lama sediment record, L.A.P. and U.H. coordinated filed work for the CH12 sediment record, I.G.A. and C.C. contributed Bolshoye Shchuchye sedaDNA extracts, K.V.K. advised nuclear gene bait design. L.S. wrote the manuscript supervised by U.H. that all co-authors commented on.

## Funding

## Competing interests

The authors declare no competing interests.
