## [Peer Review File · Communications Biology]

Reviewers' comments:

Reviewer #1 (Remarks to the Author):

The results presented in this manuscript are of relevance for people in the ancient sedimentary DNA community as well as for a broader audience. The conclusions are original and apart from two major points discussed below, the manuscript has no other flaws,

The first point of concern is related with the structure and writing style which in my opinion is not clear and should be improved before publication. Since *Communications Biology* is a non-specialistic biology journal, it is possible that non-specialists will have find difficult reading and understanding this manuscript.

The second and more important point is related with the ancient repetitive DNA region the authors present, which, hasn't been used before in *sedaDNA* analyses and it requires more clarification. The authors mention it is a "short repeat of 173 bp", which is quite long for an ancient DNA molecule, unless the repeat is assembled out of shorter reads. It is unclear if this 173 bp repeat, is the repeat motive, or if there is a smaller motive making up the 173 bp length. In the latter case, how can the authors be sure that these repeats are genus specific? Is it possible that the motives are also present in other organisms, but map to *Larix* because that's the genome used from mapping? In general, I think the manuscript would benefit having a clear description of what is the origin of the repeat and why they are specific to *Larix*, which would likely take away all the questions surrounding it. I think this is especially important since this is the first paper describing its usage in ancient sediment samples.

Other minor points

- Please add the number of lines and, at least, number of pages to make the referee work much easier.
- What exactly is the intention of using the nuclear gene capture set? Besides the bycatch of the *EulaSat1* repeat sequence, there are no specific results describing the on target reads for this capture set in the manuscript.
- Analysis of chloroplast enrichment (supporting information) – what is the length of the used kmer? This is important for reproducibility since kmer size might impact the assignment of reads.
- Although outwardly very similar, the species differ in their ecological properties and ecosystem services they offer.
- The adaptation to extreme cold prompts the question of whether the species could have survived the entire last glacial period or more specifically the Last Glacial Maximum...
- It is still under discussion whether historical factors such as the distribution of species or ecological factors such as local hydrology and habitat...
- Historical biogeography deals with long-term and large-distance patterns in distribution of species, and ecological biogeography deals with distribution of living species as a function of ecological factors like climate, latitude, precipitation etc. They are both part of the discipline biogeography; in my opinion the authors cannot use the term biogeography to mean only the historical part. The term is used in this way all over the manuscript and even in the title of a paragraph: Environment likely plays a more important role than biogeography historical factors/history

Reviewer #2 (Remarks to the Author):

This manuscript is entitled "Dynamics of larch species ranges in Siberia since the Last Glacial captured from sedimentary ancient DNA". This work highlights how a cutting-edge method, hybridization capture method applied to sedimentary DNA, to study the presence of specific species at specific times or sites. As an aquatic microbial ecologist with expertise in sedimentary

DNA research, I could not judge the importance of ecological findings related to boreal forest changes over millennia. I am instead here providing my expertise to judge the robustness of the analysis as well as helping to improve the clarity of this manuscript. Overall, I am impressed by the quality of this work. I focused my review below on improvements of the clarity of some specific aspects of the work but that can be solved by changes in the text.

First, the use of the word species to define *Larix* species is a bit confusing in the text especially from L70 where you talk about a complex for *Larix* species. From there I would guess that this includes *L. sibirica* and *L. gmelinii*. However, you refer to subspecies or two separate species but referring only to *L. gmelinii*. I think this part needs further clarifications. Please reformulate this paragraph but maybe also L119 "species". You also used the word variants L252., please harmonize.

Second, interested researchers by findings of this manuscript in the journal *Communications Biology* would not necessarily be molecular ecologists and this is for that I think there is to simplify some terms of this work. One good may be by avoiding using different words to refer to the same thing. For example, harmonize between (i) target enrichment or hybridization method. One example is L135 "enriched the samples" reads very as a shortcut of what was actually done. (ii) repetitive DNA or nuclear DNA (e.g., L146 vs L156) (iii) plastid vs chloroplast (iv) the word "reads" is used the first time L127 but never defined anywhere. I would use instead DNA sequences in the whole text because for sure, non-molecular ecologists do not like it. See for instance the title of Fig 4 "Percentage of reads"

Third, you repeat a bit too much this is the first study provided such hybridization capture data from sedaDNA from lakes in the paper such as L142, L147, L156, L291, L425. I would reduce a bit the tone especially because your previous paper (Schulte et al. 2021) already did that. Also for the statement L147, this is actually not true – see works from Tyler Murchie or Linda Armbricht for example.

Title. I think "Dynamics of larch species ranges" is a weird wording. I think spatial distributions would make more sense to describe it

L46 I would use "Siberian" instead of "Asian" to keep it consistent with your whole text.

L47 Maybe just a comment but I read it as larch species are the most dominant species in Siberian boreal forests. If this is not the case, I would reformulate.

L49 "also" in opposition to what?

L53 For readers, Holocene and Last Glacial Maximum may be not informative enough, I would put

L54 "DNA samples dated". DNA samples were not dated but sediment samples in which DNA was extracted were dated. To reformulate.

L57 I think the number of sediment samples is not necessary in the abstract

L68-71 and L83-86 feels a bit too repetitive. I would merge them.

L73 – "biogeographical split", do you mean currently or since the Last Glacial? I would clarify that.

L108-110 Is it specific to

L113 "tracing of populations in the past remains difficult as signals from different times can be superimposed". I am not sure what you mean here. I think you have to clarify what you mean by "signals" and "superimposed".

L116-117: "developed" and "develop" in the same sentence.

L131: I would be careful about using sedaDNA sample. Such sample actually do not exist because sedimentary DNA is a mix of DNA from dead and alive cells (see Capo et al. 2021 Quaternary for one – among others - definition).

L134 "a known presence of *Larix* around the lakes" That imply that you investigated past presence of *Larix* in Siberia based on the fact that if they need to be present now to also be in the past?

Sounds like a bias, right?

L137 "ancient species distribution" could be reformulated "past distribution of species" because I am not sure you are really talking about ancient species in this work (or maybe I missed something with the species complex"?)

L138 You talk about habitat and biogeography but not really climate, I would say something about climate too here because this is the start of your introduction.

L146. I have the feeling that this all section may not be very interesting for most of readers and would therefore put it in SI (just a personal feeling, not need to do it if you want it in the main text).

L172. "We raised the confidence threshold" is too vague. You could clarify that.

L203 "Even more promising than using repetitive sequence abundance in off-target capture

enrichment datasets is to use sequence variation in repeats". Either clarify what the two terms means or remove because I do not understand what are these two terms, and thus I guess like most readers of this work.

L205 "read mapping" of what?

L246 which species?

L288 remove one "of"

L289 "variable" in the sentence "pollen....are not sufficiently variable" is probably not the best word to use.

L291 "enrich the samples" is, in my opinion, meaningless, I would reformulate.

L296 : larch do not survive "at" lakes. This is a shortcut to write it this way. Please reformulate.

L302: I think mentioning the main region (e.g, Polar Ural Mountains) is better than lakes names that complexify the overall reading on this section.

L329 "L. gmelinii variations for samples" sounds meaningless too; variations of what? I would reformulate.

L477-484 = Chloroplast and l485-493 = nuclear DNA? If yes write that instead of "second line of analysis"

L485 which reads?

Good luck with revisions :)

Dear Dr. Lawing and Reviewers,

Thank you very much for handling and thorough reviewing our manuscript. We appreciate the reviewers' critique, recommendations and comments that helped us greatly improve the manuscript. We took them all into consideration while revising the manuscript and provided below for each reviewer our detailed point-by-point responses and explanations on how we addressed them. All changes are highlighted in the revised manuscript using Track Changes tool.

Best regards on behalf of all coauthors,

Luise Schulte

Reviewers' comments:

Reviewer #1 (Remarks to the Author):

The results presented in this manuscript are of relevance for people in the ancient sedimentary DNA community as well as for a broader audience. The conclusions are original and apart from two major points discussed below, the manuscript has no other flaws,

The first point of concern is related with the structure and writing style which in my opinion is not clear and should be improved before publication. Since *Communications Biology* is a non-specialistic biology journal, it is possible that non-specialists will have find difficult reading and understanding this manuscript.

Thank you for pointing this out. To make the manuscript easier to follow for people outside of the field we harmonized expressions and explained better following also the recommendations of referee 2. Amongst other things, we harmonized and changed:

- (i) harmonized the expressions "*target enrichment*", "*hybridization capture*" and "*capture enrichment*" to "*target enrichment*".
- (ii) Harmonized "*plastid*" and "*chloroplast*" to "*chloroplast*".
- (iii) Changed the word "*reads*" to "*DNA sequences*".
- (iv) Described more clearly the differences between the two datasets used and how they were analyzed, i.e. the dataset enriched for the *Larix* chloroplast and the dataset enriched for nuclear DNA of *Larix*.

The second and more important point is related with the ancient repetitive DNA region the authors present, which, hasn't been used before in sedaDNA analyses and it requires more clarification. The authors mention it is a "short repeat of 173 bp", which is quite long for an ancient DNA molecule, unless the repeat is assembled out of shorter reads. It is unclear if this 173 bp repeat, is the repeat motive, or if there is a smaller motive making up the 173 bp length. In the latter case, how can the authors be sure that these repeats are genus specific? Is it possible that the motives are also present in other organisms, but map to *Larix* because that's the genome used from mapping? In general, I think the manuscript would benefit having a clear description of what is the origin of the repeat and why they are specific to *Larix*, which would likely take away all the questions surrounding it. I think this is especially important since this is the first paper describing its usage in ancient sediment samples.

Thank you for showing the need for clarification. The repeat motif is 173 bp long. In modern larches, this 173 bp motif is arranged in tandem. Summarizing all of the repeat instances across larch genomes, this repeat makes up about 2.5% of *Larix* genomes, depending on the individual species. Given its high abundance, we retrieved many sequence reads corresponding to the 173 bp motif and assembled it with high coverage. We are basing our analysis on the repeat characterization that we have conducted prior to this manuscript, with two of the authors having been involved in the identification of repetitive sequences in larch genomes [32]. As we have tested a wide range of conifers for the presence of this repeat, we are sure that it is exclusively present in the genus *Larix*. This genus-specificity is also a common property of this type of repeat (satellite DNA). We now cite our prior study prominently at the first mention of this repeat.

In addition to our analyses that build on our prior repeat characterization, we want to make clear that for this manuscript, we only use sequences from our dataset which were taxonomically classified as *Larix*. We changed the text to make this clearer:

L181-191: *“A closer inspection of unmapped sequences assigned to Larix revealed a high content of repetitive DNAs. More specifically, taxonomically classified Larix sequences could be assembled to EulaSat1, the most abundant satellite repeat in the nuclear genome of Larix^{32,37}. This short repeat with a 173 bp long motif is arranged in large arrays of tandemly repeated motifs and is exclusively present in larches³². Analysis of modern L. sibirica, and L. gmelinii (western and eastern range) genomes reveals that EulaSat1 occurs in all species, contributing to 0.62% (L. sibirica), 2.52% (western range L. gmelinii), and 2.39% (eastern range L. gmelinii), of the genomes, respectively (Supplementary Fig. 2). A comparison of the sequence proportions mapping to the repeat motif in the different datasets of Lake CH12 showed a specific enrichment of the repeat motif by the nuclear gene hybridization probe set (Supplementary Fig. 3).”*

The origin of the satellite repeat and why it is specific to *Larix* is also shortly touched in the sentences:

L125: *“Among repetitive elements, satellite DNA repeats are among the fastest evolving and are hence often specific for a certain species or genus³²”*
and

L200: *“The main satellite in all larches, EulaSat1, is believed to have greatly multiplied after the split of Larix from Pseudotsuga³².”*

We feel that a further explanation of satellite repeat origin and evolution is out of the scope of this manuscript and refer to our prior study [32].

Other minor points

- Please add the number of lines and, at least, number of pages to make the referee work much easier.

We are very sorry that it was not visible earlier, we had somehow read that they would be generated automatically. We included them now!

- What exactly is the intention of using the nuclear gene capture set? Besides the bycatch of the EulaSat1 repeat sequence, there are no specific results describing the on target reads for this capture set in the manuscript.

It is true that the nuclear gene capture dataset does not confer any additional information which is not covered by the chloroplast capture dataset. However, we could show that the unintended bycatch of the satellite reflects the abundance patterns of the other proxies very well. Although we did not detect any sequence variation in our satellite repeat data, we could show that satellite repeats could be potentially valuable multi-copy targets for future sedaDNA studies. To explain this

better in the text we changed and added the following:

L208-216: *“Another study showed that also sequence similarities between homologous repeat motifs can be used to reconstruct phylogenetic relationships among closely related taxa^{40,41}. In the case of Larix satellite EuLaSat1 in our study, no change in nucleotide frequencies, neither related to locations nor in time, could be detected. However, our results show that the off-target fraction in target enriched sedaDNA datasets can hold valuable information and that repeat motifs in more diverse taxon groups could even be a target for enrichment. Specifically enriching for repeat motifs in sedaDNA extracts could enable the study of satellite repeat evolution as well as giving additional information on species abundance and phylogeography.”*

- Analysis of chloroplast enrichment (supporting information) – what is the length of the used kmer? This is important for reproducibility since kmer size might impact the assignment of reads.

We added a sentence in supplemental information (note that kmer size in kraken is defined in the database building step):

“The database was built using kraken2 standard parameters (kmer length of 35 bp and minimizer length of 31 bp).”

- Although outwardly very similar, the species differ in their ecological properties and ecosystem services they offer.

Agreed and added: **L77:** *“...they offer”*

- The adaptation to extreme cold prompts the question of whether the species could have survived the entire last glacial period or more specifically the Last Glacial Maximum...

Changed **L90:** *“last glacial and”* to *“entire last glacial period or”*

- It is still under discussion whether historical factors such as the distribution of species or ecological factors such as local hydrology and habitat...

- Historical biogeography deals with long-term and large-distance patterns in distribution of species, and ecological biogeography deals with distribution of living species as a function of ecological factors like climate, latitude, precipitation etc. They are both part of the discipline biogeography; in my opinion the authors cannot use the term biogeography to mean only the historical part. The term is used in this way all over the manuscript and even in the title of a paragraph: Environment likely plays a more important role than biogeography historical factors/history

Thank you for pointing this out. We agree and changed all occurrences of *“biogeography”* to *“historical factors”*

Additional changes:

L82-85: *“It is still under discussion whether historical factors such as the historical distribution of species and dispersal limitations or environmental factors such as local hydrology and habitat are more restrictive for current species distribution.”*

L390-393: *“The current boundaries of boreal Larix species arranged from west to east suggest a possible strong influence of the historical species distribution on the current distribution, whereas the gradient of increasing continental climate towards the east assumes a strong influence of environment.”*

Reviewer #2 (Remarks to the Author):

This manuscript is entitled “Dynamics of larch species ranges in Siberia since the Last Glacial captured

from sedimentary ancient DNA". This work highlights how a cutting-edge method, hybridization capture method applied to sedimentary DNA, to study the presence of specific species at specific times or sites. As an aquatic microbial ecologist with expertise in sedimentary DNA research, I could not judge the importance of ecological findings related to boreal forest changes over millennia. I am instead here providing my expertise to judge the robustness of the analysis as well as helping to improve the clarity of this manuscript. Overall, I am impressed by the quality of this work. I focused my review below how improvements of the clarity of some specific aspects of the work but that can be solved by changes in the text.

First, the use of the word species to define Larix species is a bit confusing in the text especially from l70 where you talk about a complex for Larix species. From there I would guess that this include L. sibirica and L. gmelinii. However, you refer to subspecies or two separate species but referring only to L. gmelinii. I think this part need further clarifications. Please reformulate this paragraph but maybe also l119 "species". You also used the word variants l252., please harmonize. Thank you for this valuable comment, as our description from an earlier version was indeed not clear enough. We improved this now:

L72: *"Siberian boreal Larix species are currently geographically split in a western and an eastern species (Fig. 1). West Siberia, from the western Urals to the West Siberian Plain, is dominated by Larix sibirica. Central and eastern Siberia is dominated by L. gmelinii. This species is sometimes again split into a western and eastern variety, subspecies or two separate species⁵, but officially recognized only as one species, L. gmelinii."*

L265: *"The rise in the L. sibirica DNA sequence variants coincides with a peak in read counts for Lake Kyutyunda."*

Second, interested researchers by findings of this manuscript in the journal Communications Biology would not necessary be molecular ecologists and this is for that I think there is to simplify some terms of this work. One good may be by avoiding using different words to refer to the same thing. Thank you very much for pointing this out. We revised the manuscript and changed the expressions as described below (see also the list in our answer and to Reviewer 1).

For example, harmonize between

(i) target enrichment or hybridization method. One example is l135 "enriched the samples" reads very as a shortcut of what was actually done.

Harmonized to *"target enrichment"*

(ii) repetitive DNA or nuclear DNA (e.g., l146 vs l156)

Added *"nuclear"* at some expression to make more clear that the DNA repeat is part of the nuclear genome and got enriched by the nuclear gene bait set.

L147: *"Chloroplast and repetitive nuclear DNA enrichment in the sedaDNA extracts"*

L181: *"A closer inspection of unmapped sequences assigned to Larix revealed a high content of repetitive DNAs, in particular EulaSat1, the most abundant satellite repeat in the nuclear genome of Larix"*

(iii) plastid vs chloroplast

Harmonized to *"chloroplast"*

(iv) the word "reads" is used the first time l127 but never defined anywhere. I would used instead DNA sequences in the whole text because for sure, non-molecular ecologists do not like it. See for instance the title of Fig 4 "Percentage of reads"

Harmonized to *"DNA sequences" or "sequences"*

Third, you repeat a bit too much this is the first study provided such hybridization capture data from

sedaDNA from lakes in the paper such as l142, l147, l156, l291, l425. I would reduce a bit the tone especially because your previous paper (Schulte et al. 2021) already did that. Also for the statement l147, this is actually not true – see works from Tyler Murchie or Linda Armbrrecht for example.

We agree and deleted L158 and changed L147 to state that it is the first large scale capture dataset from lake sediments (our previous study was a pilot study with only 4 samples from one lake, reaching back only 7000 years).

L149: *“We generated the first large-scale target enriched dataset using sedaDNA extracted from lake sediments.”*

Title. I think “Dynamics of larch species ranges” is a weird wording. I think spatial distributions would make more sense to describe it

We changed the title to „Larix species range dynamics in Siberia since the Last Glacial captured from sedimentary ancient DNA”

L46 I would use “Siberian” instead of “Asian” to keep it consistent with your whole text.

Agreed and changed.

L47 Maybe just a comment but I read it as larch species are the most dominant species in Siberian boreal forests. If this is not the case, I would reformulate.

Larix dominates indeed 81% of Siberian boreal forests, see also in **L65**.

L49 “also” in opposition to what?

Changed to **L48:** *“in part”*

L53 For readers, Holocene and Last Glacial Maximum may be not informative enough, I would put Agreed and changed to:

L51: *“We discovered that Larix sibirica, presently dominating in western Siberia, likely migrated to its northern distribution area only in the Holocene at around 10,000 years before present (ka BP), and had a much wider eastern distribution around 33 ka BP. Samples dated to the Last Glacial Maximum (around 21 ka BP), consistently show genotypes of L. gmelinii.”*

L54 “DNA samples dated”. DNA samples were not dated but sediment samples in which DNA was extracted were dated. To reformulate.

Agreed and changed to **L53:** *“Samples dated”*

L57 I think the number of sediment samples is not necessary in the abstract

Agreed and deleted.

L68-71 and l83-86 feels a bit too repetitive. I would merge them.

To avoid repetition we deleted the sentence:

“This knowledge however represents the basis for prediction of future forest.”

L73 – “biogeographical split”, do you mean currently or since the Last Glacial? I would clarify that.

Changed to:

L72: *“Siberian boreal Larix species are currently geographically split in a western and an eastern species”*

L108-110 Is it specific to

Sorry, parts of the comment seem to be missing but we understand what you wanted to say. To clarify the sentence in the lines mentioned before, we changed it to:

L107: *“Traditional paleoecological approaches are based on fossil pollen isolated from lake sediments, peat or permafrost sections or metabarcoding using short diagnostic metabarcodes of ancient DNA extracted from the same kind of samples.”*

L113 “tracing of populations in the past remains difficult as signals from different times can be superimposed”. I am not sure what you mean here. I think you have to clarify what you mean by “signals” and “superimposed”.

We made it more clear and changed to:

L112: “Nevertheless, tracing of populations in the past using modern genetic data remains difficult. Changes in genetic variation as a result of specific events in different times (such as population expansion or retraction) can be superimposed and only a few studies have hitherto been available to serve as ground truth estimates for eastern Siberia^{21,27,28}.”

L116-117: “developed” and “develop” in the same sentence.

Agreed and changed it to:

L116: “Recently, Schulte et al.²⁷ developed target enrichment of ancient *Larix* DNA by using the complete chloroplast genome of *Larix* as bait set (hybridization oligonucleotides) on sedimentary ancient DNA (sedaDNA).”

L131: I would be careful about using sedaDNA sample. Such sample actually do not exist because sedimentary DNA is a mix of DNA from dead and alive cells (see Capo et al. 2021 Quaternary for one – among others - definition).

Agreed and changed to:

L131: “sedaDNA extracts, isolated from sediment core samples”

L134 “a known presence of *Larix* around the lakes” That imply that you investigated past presence of *Larix* in Siberia based on the fact that if they need to be present now to also be in the past? Sounds like a bias, right?

Thank you for your concern. We meant a known presence in the past. Changed to:

L133: “The lakes were selected according to the availability of sediment records spanning the LGM and a known presence of *Larix* in the past around the lakes as indicated from other proxies from the same location such as pollen and metabarcoding”

L137 “ancient species distribution “could be reformulated “past distribution of species” because I am not sure you are really talking about ancient species in this work (or maybe I missed something with the species complex”?

Agreed and changed to **L138:** “past distribution of species”.

L138 You talk about habitat and biogeography but not really climate, I would say something about climate too here because this is the start of your introduction.

Agreed and changed to:

L139: “We show that climate and local habitat such as permafrost is a stronger determinant of species distribution than historical factors.”

L146. I have the feeling that this all section may not be very interesting for most of readers and would therefore put it in SI (just a personal feeling, not need to do it if you want it in the main text).

Thank you for bringing up your concern. We chose to leave it in the main text as we believe it is crucially important for the understanding of the data.

L172. “We raised the confidence threshold” is too vague. You could clarify that.

Agreed and added in parentheses:

L174: “a parameter defining the number of *k*-mers needed to produce a match against a taxon in the database”

L203 “Even more promising than using repetitive sequence abundance in off-target capture enrichment datasets is to use sequence variation in repeats”. Either clarify what the two terms means or remove because I do not understand what are these two terms, and thus I guess like most readers of this work.

Thank you for asking to clarify this section. We agree that it was not very well understandable. We clarified and changed this section to:

L208: *“Another study showed that also sequence similarities between homologous repeat motifs can be used to reconstruct phylogenetic relationships among closely related taxa^{39,40}. In the case of Larix satellite EuLaSat1 in our study, no change in nucleotide frequencies, neither related to locations nor in time, could be detected. However, our results show that the off-target fraction in target enriched sedaDNA datasets can hold valuable information and that repeat motifs in more diverse taxon groups could even be a target for enrichment. Specifically enriching for repeat motifs in sedaDNA extracts could enable the study of satellite evolution as well as giving additional information on species abundance and phylogeography.”*

L205 “read mapping” of what?

Changed to:

L217: *“In the two target enriched datasets, sequences taxonomically classified to the genus Larix and mapping to the chloroplast and to the repeat sequence respectively, show similar patterns of abundance”*

L246 which species?

L259: Changed “the species” to “*L. sibirica*”

L288 remove one “of”

We agree and deleted one.

L289 “variable” in the sentence “pollen....are not sufficiently variable” is probably not the best word to use.

We agree and changed to:

L302: *“The question which of the Larix species formed these populations has hitherto been unanswered, as both pollen and established metabarcoding markers are not able to distinguish between species in the genus Larix, and findings of fossilized cones identifiable to species are rare.”*

L291 “enrich the samples” is, in my opinion, meaningless, I would reformulate.

We agree and changed to:

L305: *“By enriching sedaDNA extracts for chloroplast genome sequences”*

L296 : larch do not survive “at” lakes. This is a shortcut to write it this way. Please reformulate.

Agreed and changed to:

L312: *“in the vicinity of”*

L302: I think mentioning the main region (e.g, Polar Ural Mountains) is better than lakes names that complexify the overall reading on this section.

Thank you for the comment, however, our concern is that it is easier for the reader to relate the described results with the plots which are also labeled with lake names.

L329 “L. gmelinii variations for samples” sounds meaningless too; variations of what? I would reformulate.

We agree and changed to:

L318: *“all Pleistocene samples show similarly a dominance of L. gmelinii sequence variations”*

L477-484 = Chloroplast and l485-493 = nuclear DNA? If yes write that instead of “second line of analysis”

Agreed and changed to:

L493: *“In the first dataset, where sedaDNA extracts were enriched for Larix chloroplast DNA, quality controlled DNA sequences were first classified against a chloroplast database from NCBI refseq⁶⁸ using KRAKEN2⁶⁹ with the very conservative confidence threshold of 0.8 and only sequences classified to Larix at genus or species level were used for further analysis.”*

And:

L503: *“In the second dataset, where sedaDNA extracts were enriched for a set of Larix nuclear genes, quality-controlled DNA sequences were classified with KRAKEN2 against the plant database from NCBI RefSeq.”*

L485 which reads?

Changed to:

L503: *“In the second dataset, where sedaDNA extracts were enriched for a set of Larix nuclear genes, quality-controlled DNA sequences were classified with KRAKEN2 against the plant database from NCBI RefSeq.”*

Good luck with revisions :)

Eric Capo

REVIEWERS' COMMENTS:

Reviewer #1 (Remarks to the Author):

Dear authors, I am pleased with your revised version and I congratulate with the important work done with now takes on a more modest tone and is more open for a broader audience. It remains only to improve the text in two places in my opinion where I offer the following suggestions:

Line 123: Other, yet unexplored, multi-copy target regions for enrichment, are repetitive DNA sequences (satellite repeats) in the nuclear plant genome, which occupy about 80% of...and are believed to be the main source of genome expansion.

Line 138: did you not enrich from DNA libraries, and not directly from DNA extracts?

Reviewer #2 (Remarks to the Author):

The authors satisfactorily revised this manuscript. Good job!